# Effect of a blood culture collection bundle on decreasing the contamination rate

**Miki Doi**[1], **Yoshio Takesue**[2,3]*, **Miyuki Makino**[3], **Yousuke Kihara**[3], **Akiko Tanikawa**[3], **Yasushi Murakami**[4], **Hitoshi Ogashiwa**[1], **Yukiko Nakano**[1], **Soichiro Nakama**[3], **Takashi Ueda**[2], **Kazuhiko Nakajima**[2], **Yasuhiro Nozaki**[4]

1 Department of Clinical Technology, Tokoname City Hospital, Tokoname, Aichi, Japan, 2 Department of Infection Prevention and Control, Hyogo Medical University Hospital, Nishinomiya, Hyogo, Japan, 3 Department of Infection Prevention and Control, Tokoname City Hospital, Tokoname, Aichi, Japan, 4 Department of Respiratory Medicine, Tokoname City Hospital, Tokoname, Aichi, Japan

* takesuey@hyo-med.ac.jp

**Data Availability Statement:** All relevant data are within the paper and its Supporting information files.

**Funding:** The author(s) received no specific funding for this work.

## Abstract

In this study, we examined the effect of a bundled approach to blood collection for blood culture on decreasing contamination. Commensal organisms were considered contaminants on the basis of the clinical course if they were recovered from only a single blood draw (set) and if a positive result for two sets was not obtained within 72 hours. The main elements of the bundle were blood collection by venipuncture, skin preparation with a chlorhexidine alcohol swab, disinfection of culture bottles, and use of a sterile blood transfer device instead of the two-needle technique for inoculation. In the bundle intervention, chlorhexidine alcohol was first introduced in the hospital, and use of the blood transfer device was increased during the intervention. Both items were used in most patients requiring blood cultures. Blood collection through a line caused contamination in only one case. The contamination rate decreased significantly from 2.0% to 1.0% after introduction of the bundle approach (3-year control period vs. 2-year bundle period, p<0.001), and a significant decrease in the contamination rate was observed for coagulase-negative *Staphylococcus* (p<0.001). A high contamination rate was found in August and September during the control period. The contamination rate of *Bacillus* species was significantly higher in those 2 months than in other months. A seasonal increase was not observed during the bundle period. A low contamination rate of 1.0% was achieved using our bundled collection approach for blood culture.

## Introduction

The identification of true pathogens and subsequent results regarding antibiotic susceptibility are crucial information to provide optimal treatment for blood stream infections [1]. Inadequate collection of blood culture (BC) specimens is associated with suboptimal therapy, leading to poor clinical outcomes owing to false-negative BC or false-positive BC (contamination) results, which might also cause an increased financial burden and prolonged hospital stay [2–5].

**Competing interests:** The authors have declared that no competing interests exist.

Common BC contaminants are coagulase-negative *Staphylococcus* (CNS), *Bacillus* spp., *Corynebacterium* spp., *Micrococcus* spp., as well as *Cutibacterium acnes* and related species [6]. When any of these organisms are recovered from a single BC, careful assessment is required to determine their clinical importance. Alpha-hemolytic *Streptococcus* and *Clostridium* spp. have varying levels of clinical importance as these microorganisms could be contaminants or true pathogens.

Potential causes for a false-negative BC include an inadequate blood collection volume, an insufficient number of sets collected, and collection of samples after starting antibiotic therapy. The potential causes of BC contamination include inadequate skin antisepsis and inoculation without disinfection of BC bottles [7–10]. Collection of blood specimens through existing intravascular catheters is associated with a higher rate of BC contamination than collection using peripheral venipuncture [11]. The contamination rate is also higher with optional sterile gloving after disinfection of a skin venipuncture site than routine sterile gloving [12, 13].

Blood collection into culture bottles is done with venipuncture in countries where BC is classified as an in vitro diagnostic procedure. However, direct blood collection into culture bottles is prohibited in Japan because BC is not approved as an in vitro diagnostic method. Therefore, in Japan, blood collected in the syringe is transferred to culture bottles with use of a blood transfer device after the completion of venipuncture (S1 Fig). Although this device was introduced to prevent needle stick injury during inoculation of the culture bottle, use of a sterile blood transfer device for the inoculation of culture bottles, versus a needle that is potentially contaminated with skin microorganisms, would decrease the risk of BC contamination.

Previous benchmark for the incidence of BC contamination was typically ≤3.0% [14, 15] and further decrease is suggested. An antimicrobial stewardship team should validate the process of BC and minimize the contamination rate to less than the upper threshold of the benchmark. We updated our protocol and developed a bundled approach to perform appropriate collection of blood samples for BC. In this study, we aimed to determine whether this blood culture collection bundle could decrease the BC contamination rate.

## Methods

### Study design

This study was conducted at Tokoname City Hospital, a community hospital with 266 beds that has no intensive care unit. Patients with multiple BC sets collected on the same day were included in the study. This study included the period between May 2019 and April 2022 in which a previous protocol (S1 Table) for collecting blood samples for BC was followed (control period) and a period between May 2022 and April 2024 in which the developed blood collection bundle intervention was followed (bundle period). This study was approved by the Institutional Review Board of our hospital (No. 2023–7). The board waived the requirement for obtaining informed consent from patients who were included in this study. The opt-out approach was used. The authors collected data from patients' medical records between December 1, 2023 and June 1, 2024. We used de-identified patient data and the authors did not access any information that could be used to identify individual patients after data collection.

### Bacterial culture method

The BD BACTEC® FX system (Nippon Becton, Dickinson Company, Ltd., Tokyo, Japan) was used for automatic BC analysis. Blood samples were inoculated into one aerobic bottle and one anaerobic bottle (BD BACTEC Plus aerobic and Lytic anaerobic media, respectively). AccuRate® Separated Sheep Blood Agar/BTB Lactose Agar (Shimadzu Diagnostics

Corporation, Tokyo, Japan) or TSAII 5% sheep blood agar/chocolate II agar 100 sheets (Nippon Becton, Dickinson Company, Ltd.) were used for culture media.

## Definition of BC contamination

Bacteria associated with contaminated BCs included CNS, alpha-hemolytic *Streptococcus* except for *Streptococcus pneumoniae*, *Streptococcus gallolyticus*, *Streptococcus anginosus*, *Corynebacterium* spp. and related genera, *Bacillus* spp. other than *Bacillus anthracis*, *Micrococcus* spp., *Cutibacterium acnes* and related species, *Clostridium* spp. such as *Clostridium perfringens*, organisms that were previously classified as *Peptostreptococcus* spp., and environment-related bacteria.

The above organisms were considered contaminants if they were recovered only from a single BC set among multiple sets collected on the same day and if they were positive in BC in more than a single set at >72 hours of incubation. The final decision regarding BC contamination was made by a physician from the Infectious Disease Department on the basis of the clinical course. Retrospective assessment by the physician was performed before April 2021. However, assessment was also made in an antimicrobial stewardship meeting after this time. The physician from the Infectious Disease Department suggested an additional diagnostic imaging test if necessary to exclude infectious endocarditis and vertebral osteomyelitis. Blood collection through an intravascular catheter or from the femoral site was taken into account in the assessment of BC contamination. The following microorganisms were considered true pathogens and not contaminants even if positive in a BC in a single set: *Staphylococcus aureus*; Group A streptococci; *Streptococcus pneumoniae*; *Pseudomonas aeruginosa*; *Enterobacteriaceae*; *Bacteroidaceae*; *Haemophilus influenzae*; and *Candida* spp. The final decision was also made by the physician from the Infectious Disease Department.

## Bundle intervention for collecting blood samples for BC

The protocol before blood collection was as follows. BCs were obtained before starting antimicrobial therapy. Hand hygiene was carried out using an alcohol-based hand rub. Sterile gloves were worn if re-palpating the vein was anticipated because of presumed difficulty with blood collection.

The bundled approach for collecting blood samples for BC was as follows.

1. For specimen collection, blood samples were obtained via venipuncture rather than through an intravascular catheter. A catheter draw along with a simultaneous peripheral draw was indicated when a catheter-related blood stream infection was suspected.

2. To prepare the venipuncture site, the skin at the collection site was scrubbed with an alcohol swab. The skin was then disinfected with a 1% chlorhexidine alcohol swab using repeated back-and-forth strokes. A drying time of 30 seconds was required to obtain good efficacy.

3. The rubber stopper of the BC bottle was disinfected with an alcohol swab.

4. The volume of blood collection was 16–20 mL for each draw (8–10 mL each for aerobic and anaerobic bottles).

5. Two or more separate blood draws (sets) on the same day were required.

6. A sterile blood transfer device was used. The needle that was used for venipuncture was changed for transferring blood to culture bottles. When a straight needle was detached, a

needle-removing device on the sharps disposal container was used (no risk of contamination during the procedure was ensured as described in S2 Table).

7. If less than the recommended volume of blood was drawn, the aerobic bottle was first inoculated to the required fill mark, followed by transfer of the remaining blood to the anaerobic bottle. Conventionally, the blood was first inoculated into the anaerobic bottle because of oxygen sensitivity in obligate anaerobes. However, most organism growth was recovered from aerobic bottles. Therefore, if the volume of blood collected was <10 mL, the entire sample was inoculated into the aerobic bottle.

### Evaluation of adherence to the BC bundle

Adherence to the main components of the bundle was evaluated using two different methods. Achievement of culture bottle disinfection, disinfection of the venipuncture site, and use of a sterile blood transfer device was monitored on random occasions by link nurses in each ward between July 2022 and April 2024. Additionally, institutional use of skin disinfection swab sets and sterile blood transfer devices was evaluated.

### Evaluation of the BC contamination rate

The contamination rate was generally calculated by dividing the total number of contaminated BC sets by the total number of BC sets collected during the evaluation period. The contamination rate per set is the optimal marker for tracking the BC contamination rate at an institution or to compare the contamination rate per set between institutions. However, to evaluate the efficacy of bundle management, dividing the total number of patients with BC contamination (events) by the total number of patients in whom BC was conducted (events) is the best marker. A bundle was generally prepared to manage individual patients, but not for each blood sample collection set. An example of why this approach was chosen is that if one set grows *Staphylococcus epidermidis* and the patient's other set is negative, then implementation of the bundled approach would lead to judgment of a failed BC in this patient. Therefore, in this study, the BC contamination rate was evaluated for both methods (i.e., each set and each event) according to the purpose of use.

### Statistical analyses

Continuous variables are presented as mean ± standard deviation if the data followed a normal distribution. The median (interquartile range [IQR]) was used for analysis if the data were skewed because the mean value could be distorted by outliers. Parametric variables were analyzed using the Student *t*-test whereas nonparametric variables were analyzed using the Mann–Whitney *U*-test. The level of significance was set at p<0.05. IBM SPSS ver. 24 (IBM Corp., Armonk, NY, USA) was used to perform the analyses.

## Results

Among BCs, there were 3111 events during the control period and 2296 events during the bundle period. After excluding events in which only one set was obtained, 2858 events and 2176 events were included in the study, respectively. Similar age, sex, rate of blood collection for BC set in the emergency outpatient clinic and BC-related factors were found between the two study periods, including the number of BC sets/1000 patient days, central venous line utility rate, and incidence of true bacteremia or fungemia (Table 1). During both periods, the

**Table 1. Patient background and blood culture-related factors before and after revision of the blood culture collection protocol.**

| Blood culture-related factors | Control period | Bundle period |
|---|---|---|
| No. of events included in the study | 2858 | 2176 |
| Median patient age (interquartile range) | 80 years (71–86) | 81 years (74–87) |
| Sex, male | 55.2% | 55.1% |
| No. of sets included in the study (/1000 patient days) | 5828 (30.5) | 4359 (35.7) |
| No. of sets obtained in emergency outpatient clinic (rate) | 2624 (45.8%) | 2072 (47.5%) |
| Device utility rate for a central venous line | 0.037 | 0.04 |
| True bacteremia or fungemia, median no. of events/month (interquartile range) | 11.0 (9.0–13.0) | 11.5 (8.8–14.0) |
| Organisms isolated from patients with true bacteremia or fungemia, n (%) | | |
| *Escherichia coli* | 138 (30.2) | 84 (28.7) |
| *Staphylococcus aureus* | 57 (12.5) | 50 (17.1) |
| *Streptococcus* spp. | 55 (12.0) | 32 (10.6) |
| *Klebsiella pneumoniae* | 40 (8.8) | 27 (9.2) |
| *Enterococcus* spp. | 32 (7.0) | 25 (8.5) |
| Anaerobes | 17 (3.7) | 20 (6.8) |
| Coagulase-negative *Staphylococcus* | 32 (7.0) | 11 (3.8) |
| *Bacillus* spp. *(B. cereus)* | 7 (1.5) (6) | 2 (0.7) (1) |
| *Candida* spp. | 10 (2.2) | 8 (2.7) |
| Other fungi | 2 (0.5) | 0 |
| Other | 67 (14.7) | 35 (11.9) |

most common isolate involved in true bacteremia or fungemia was *Escherichia coli*, followed by *S. aureus* and *Streptococcus* spp. The proportion of CNS was only 7.0% in the control period and 3.8% in the bundle period. In this study, the lower central line utility rate (0.047) and higher incidence of secondary bacteremia following community-acquired infection (organisms were isolated from specimen collected at the emergency outpatient clinic before admission in 56.1% of patients with bacteremia/fungemia) caused the particular isolation pattern of the organisms involved in true bacteremia, both factors of which are typical characteristics in community hospitals without an intensive care unit as compared with tertiary care hospitals.

The main isolate causing BC contamination was CNS (94 strains), followed by *Bacillus* spp. (22 strains). The proportion involved in BC contamination was approximately 70% for CNS, *Bacillus* spp., and *Corynebacterium* spp. and that for alpha-hemolytic *Streptococcus* was 47.4% (Table 2). Reasons for the determination of BC contamination are shown in S3 Table. Isolation

**Table 2. Organisms causing bacterial culture contamination and the contamination rate for each organism during the study period.**

| Common contaminants | Rate among contaminants (n = 161) | Contamination rate in each organism (per set) |
|---|---|---|
| Coagulase-negative *Staphylococcus* | 58.4% | 68.6% (94/137) |
| *Bacillus* spp. | 13.7% | 71.0% (22/31) |
| *Corynebacterium* spp. | 5.6% | 75.0% (9/12) |
| Alpha-hemolytic *Streptococcus* | 5.6% | 47.4% (9/19) |
| *Cutibacterium* spp., *Propionibacterium* spp. (including *C. acnes*) | 3.1% | 100% (5/5) |
| *Clostridium perfringens* | 3.1% | 62.5% (5/8) |
| *Clostridium* spp. except for *C. perfringens* | 1.9% | 42.9% (3/7) |
| Other anaerobes | 5.6% | 75.0% (9/12) |

from only a single set was the reason for contamination in most organisms. In four organisms with a positive result in two sets, a duration of incubation time ≥72 hours or blood sampling through the line explained the BC contamination.

In the control period, a povidone-iodine swab stick (Povidone Plus®, Kawamoto Corporation, Osaka, Japan) was exclusively used for skin preparation before BC blood collection. Chlorhexidine alcohol swabs were introduced for the first time in May 2022 for bundle implementation (Fig 1).

The Hexizac AL 1% Cotton Stick® (Yoshida Pharmaceutical Company Limited, Tokyo, Japan) or a 1% chlorhexidine gluconate ethanol solution® (Kenzmedico Co., Ltd., Saitama, Japan) were also used. As for median monthly use, chlorhexidine alcohol swabs were used in 408 sets (IQR 288–519) between May 2022 and April 2044 (bundle period). With the change in skin disinfectant, use of the povidone-iodine stick gradually decreased between the control period and the bundle period (median: 795 sets [IQR 720–1080] vs. 510 sets [IQR 420–630], p<0.001). The skin disinfection swab set included two sticks in povidone-iodine, which were used in a concentric circle fashion from the central area to the outer edge, and one stick in 1% chlorhexidine alcohol, which was used in repeated back-and-forth strokes.

A 1% chlorhexidine alcohol swab stick was also used for skin disinfection in patients with an indwelling central venous catheter and insertion of a thoracostomy tube. Following the expanded indication in September 2023 for the 1% chlorhexidine alcohol stick in skin preparation among patients receiving hemodialysis, the institutional monthly use increased. Concomitant with the swab stick, 1% chlorhexidine gluconate ethanol solution (250 mL) (Yoshida Pharmaceutical Company Limited) has been used for the preparation of angiography and interventional radiology as well as some surgeries since January 2024.

Sterile blood transfer devices were introduced in December 2021 to transfer venous blood from a syringe to an evacuated blood collection tube for biochemical and peripheral blood cell examination. The BD blood transfer device (Vacutainer®; Nippon Becton Dickinson Company, Ltd., Tokyo, Japan) or Greiner blood transfer unit (VACUETTE®; Greiner Japan. Ltd., Tokyo, Japan) was used in this study.

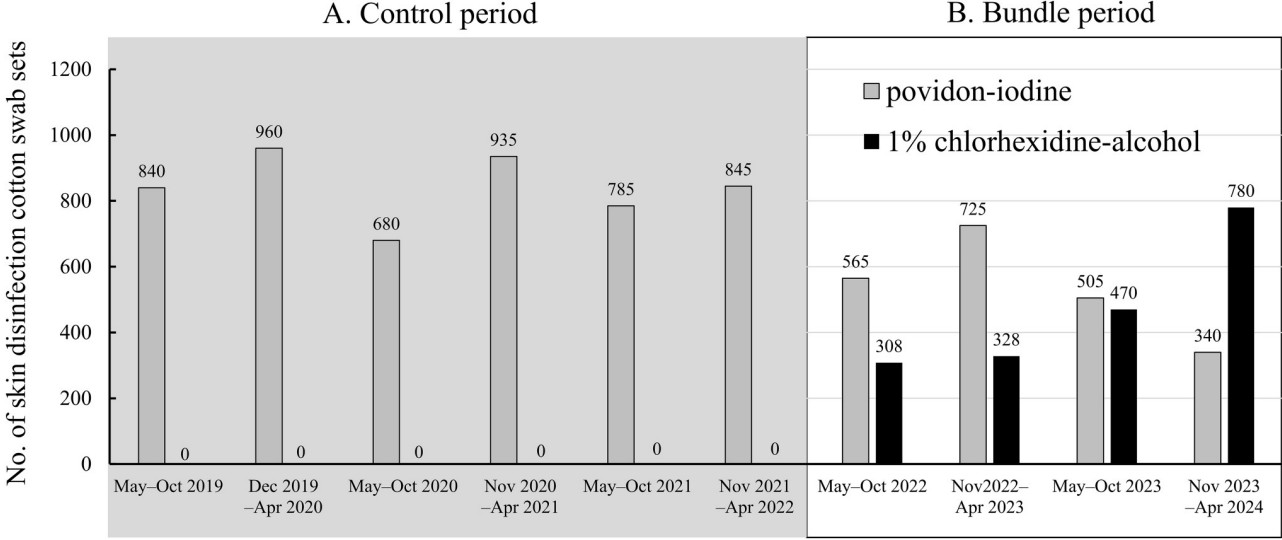

**Fig 1. Semiannual analysis of monthly use of skin disinfection swab sets in the hospital.** A. May 2019 to April 2022; B. May 2022 to April 2024. Gray bar chart: povidone-iodine, Black bar chart:1% chlorhexidine-alcohol.

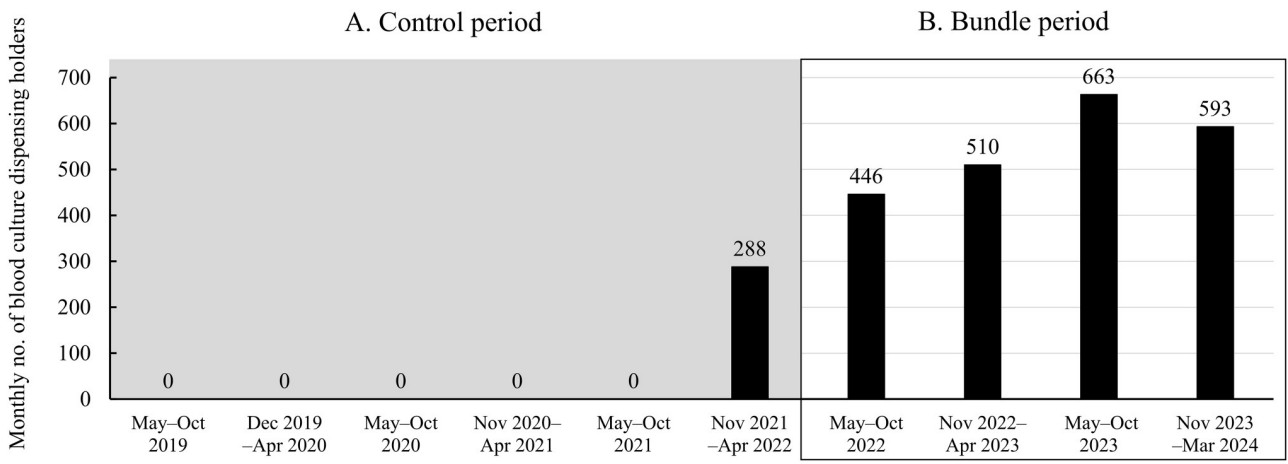

**Fig 2. Semiannual analysis of monthly use of sterile blood transfer devices.** A. May 2019 to April 2022; B. May 2022 to April 2024.

During the control period, use of the device was prohibited for transferring blood from a syringe to a BC bottle, and a conventional single-needle transfer technique was adopted. Use of the device to transfer blood to culture bottles was started following the introduction of the bundle, and institutional median monthly use significantly increased to 585 kits (IQR 424.5–646.5) in the bundle period (May 2022 to April 2024) comparing with that in the previous 6 months (p = 0.008) (Fig 2).

A random survey for achieving the main elements of the bundle was conducted on 429 occasions. Disinfection of the top of the culture bottle was carried out in all events. A 1% chlorhexidine alcohol swab stick was used in 427 (99.5%) events and a sterile blood transfer device was used in 424 (98.8%) events.

Among contaminated BCs, blood was collected in the emergency outpatient clinic in 66 patients (60.6%) during the control period and in 26 of 40 patients (65%) during the bundle period. Seventeen of 109 patients with BC contamination in the control period and 5 of 40 patients with BC contamination in the bundle period had a central venous catheter when venipuncture was conducted for BC; blood was collected through the line in 0 of 17 patients with contaminated BCs during the control period and in 1 of 5 patients with contaminated BCs during the bundle period. During both periods, no blood was drawn through an arterial line in any patients with BC contamination.

The contamination rate per set decreased significantly from 2.0% in the control period to 1.0% in the bundle period (p<0.001, Table 3). The contamination rate per event also decreased significantly in the bundle period compared with that in the control period (p<0.001). The contamination rate in positive BCs decreased significantly with introduction of the bundle (per set: p = 0.008; per event: p = 0.005). The contamination rate of CNS per set was

**Table 3. Comparison of the blood culture contamination rate between the control and bundle periods according to the evaluation method.**

| Evaluation method of the contamination rate | All blood cultures | | | Positive blood culture | | |
|---|---|---|---|---|---|---|
| | Control period | Bundle period | p value | Control period | Bundle period | p value |
| Rate per set | 2.0% (112/5728) | 1.0% (44/4359) | <0.001 | 13.7% (112/820) | 8.8% (44/500) | 0.008 |
| Rate per event | 3.8% (109/2858) | 1.8% (40/2176) | <0.001 | 21.2% (109/513) | 13.4% (40/299) | 0.005 |

significantly decreased in the bundle period compared with that in the control period (p<0.001). We observed a decreasing trend in the contamination rate of *Bacillus* spp. during the bundle period compared with the findings in the control period (p = 0.084) (Table 4). The proportion of CNS among contaminants was significantly decreased in the bundle period compared with that in the control period (p = 0.015) whereas the proportion of anaerobes was significantly increased (p = 0.017) (S4 Table).

The change in the monthly contamination rate (per set) was evaluated. A marked increase in the contamination rate was observed in August 2020 and also in August and September 2021 (Fig 3).

During the summer season (August and September) in the control period, the most common isolate was CNS (24 strains), followed by *Bacillus* spp. (*B. cereus*, five strains; and *B. subtilis*, three strains). There was a significantly higher contamination rate of *Bacillus* spp. in these 2 months than in the other 10 months (0.84% [8 strains/948 sets] vs. 0.19% [9 strains/4780 sets], p = 0.002) in the control period. The contamination rate of CNS in these 2 months was also

**Table 4. Comparison of the blood culture contamination rate (per set) in each organism between the control and bundle periods.**

| Contaminant | Control period (n = 5728) | Bundle period (n = 4359) | p value |
|---|---|---|---|
| Coagulase-negative *Staphylococcus* | 74 strains (1.29%) | 20 strains (0.46%) | <0.001 |
| *Staphylococcus epidermidis* | 32 | 6 | |
| *Staphylococcus capitis* | 10 | 7 | |
| *Staphylococcus hominis* | 8 | 2 | |
| *Staphylococcus caprae* | 4 | 3 | |
| *Staphylococcus simulans* | 5 | 0 | |
| *Staphylococcus pettenkoferi* | 4 | 0 | |
| Other | 11 | 2 | |
| *Bacillus* spp. | 17 strains (0.30%) | 5 strains (0.11%) | 0.084 |
| *Bacillus cereus* | 8 | 1 | |
| *Bacillus subtilis* | 8 | 2 | |
| Other | 1 | 2 | |
| *Corynebacterium* spp. | 6 strains (0.10%) | 3 strains (0.07%) | 0.793 |
| Alpha-hemolytic *Streptococcus* | 4 strains (0.07%) | 5 strains (0.11%) | 0.681 |
| *Micrococcus* spp. | 2 strains (0.035%) | 0 strain | – |
| Anaerobes | 11 strains (0.19%) | 11 strains (0.25%) | 0.520 |
| *Clostridium perfringens* | 3 | 2 | |
| *Clostridium clostridioforme* | 1 | 0 | |
| *Clostridium paraputrificum* | 0 | 1 | |
| *Terrisporobacter glycolicus* | 0 | 1 | |
| *Cutibacterium acnes* | 2 | 1 | |
| *Propionibacterium* spp. | 1 | 1 | |
| *Anaerococcus prevotii* | 1 | 0 | |
| *Finegoldia magna* | 1 | 2 | |
| *Parvimonas micra* | 1 | 1 | |
| *Actinomyces naeslundii* | 1 | 0 | |
| *Actinomyces meyeri* | 0 | 1 | |
| *Eubacterium limosum* | 0 | 1 | |
| Organisms isolated from the environment | 1 strain (0.017%) | 2 strains (0.046%) | – |
| *Moraxella* spp. | 1 | 0 | |
| *Burkholderia cepacia* | 0 | 1 | |
| *Rothia* spp. | 0 | 1 | |

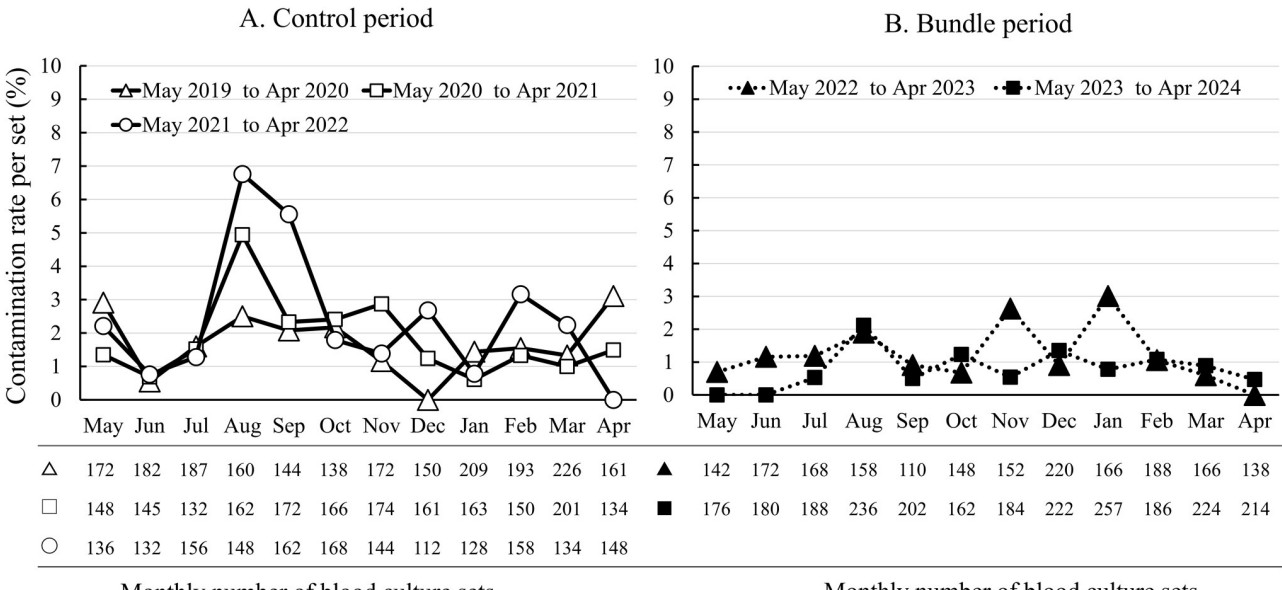

**Fig 3. Changes in the monthly contamination rate (per set) in blood culture (BC).** A. Control period, Open triangles solid line: May 2019 to Apr 2020, Open square solid line: May 2020 to Apr 2021, Open circles solid line: May 2021 to Apr 2022. B. Bundle period, Solid triangles dotted line: May 2022 to Apr 2023, Solid circles dotted line May 2023 to Apr 2024.

significantly higher than that in the other 10 months during the control period (2.53% [24 strains/948 sets] vs. 1.05% [50 strains/4780 sets], p<0.001). In contrast, a seasonal increase in the contamination rate was not observed in the bundle period. During August and September, the BC contamination rate in the bundle period was significantly lower than that in the control period (1.73% vs. 4.01%, p = 0.007). The effectiveness of the bundle intervention was demonstrated not only in these 2 months but also in the other months, and there was a significantly lower BC contamination rate in the bundle period compared with the rate in the control period during the 10 months from October to July (0.93% vs. 1.64%, p = 0.007) (Table 5).

In addition to BC contamination, four strains of *Bacillus* sp. (*B. cereus*, three strains) were isolated from patients with true bacteremia in August and September, as compared with three strains (*B. cereus* three strains) in the other 10 months during the control period. The number of *B. cereus* strains involved in contamination or true bacteremia was three in 2019 (isolated from two wards), four in 2020 (from an outpatient clinic before patient admission [n = 3] and from one ward [n = 1]), and four in 2021 (from an outpatient clinic before patient admission [n = 2] and from two wards [n = 2]). No apparent nosocomial transmission on the same ward was observed.

**Table 5. Comparison of the contaminant isolation rate between the control and bundle periods in August and September and in the remaining 10 months.**

| | No. of contaminants (rate among total sets) in August and September | | | No. of contaminants (rate among total sets) in the remaining 10 months | | |
|---|---|---|---|---|---|---|
| | Control period (n = 948) | Bundle period (n = 706) | p value | Control period (n = 4780) | Bundle period (n = 3653) | p value |
| Coagulase-negative *Staphylococcus* | 24 (2.53%) | 7 (0.99%) | 0.036 | 50 (1.05%) | 13 (0.36%) | <0.001 |
| *Bacillus* spp. | 8 (0.84%) | 1 (0.14%) | 0.114 | 9 (0.19%) | 4 (0.11%) | 0.526 |
| Other | 6 (0.63%) | 4 (0.57%) | 0.882 | 18 (0.38%) | 17 (0.47%) | 0.530 |
| Total | 38 (4.01%) | 12 (1.70%) | 0.007 | 77 (1.61%) | 34 (0.93%) | 0.007 |

## Discussion

A target performance standard rate of <3% was previously recommended for BC contamination. An even lower rate of ≤1% is now suggested as a target because many facilities have already achieved this benchmark [16, 17]. With the implementation of bundles based on novel evidence, a BC contamination rate of 1.0% was achieved in this study. The main elements in the bundle for preventing BC contamination were skin disinfection with chlorhexidine alcohol, disinfection of the top of the BC bottle [18], use of a sterile blood transfer device instead of the two-needle technique for transferring collected blood to culture bottles, and not drawing blood through an intravascular line unless a catheter-related blood stream infection is suspected. A high achievement of each element was confirmed with the institutional use of chlorhexidine alcohol and sterile blood transfer devices, as well as bedside monitoring in a random fashion. Additionally, blood collection through a line was performed in only one patient with BC contamination during the bundle period.

Several authors have reported a lower contamination rate with skin preparation using chlorhexidine alcohol than using povidone-iodine [19, 20]. Following the removal of skin contamination with an alcohol swab, disinfection using repeated back-and-forth strokes is recommended when using chlorhexidine alcohol versus disinfection with aqueous-based povidone-iodine using concentric circles from the center to the outer edge. In these two disinfection procedures, two swab sticks are required when using povidone-iodine for a venipuncture site whereas skin preparation can be performed with one stick when using chlorhexidine alcohol.

Discarding the needle used for venipuncture and switching to a new needle to inoculate culture bottles was previously standard practice. A meta-analysis [21] showed a lower contamination rate using the double-needle technique than using the single-needle technique. However, the double-needle technique is currently discouraged because of the risk of needle stick injury; instead, using the same needle to draw blood and to inoculate the culture bottles is recommended [7, 8]. However, changing from a needle to a sterile blood transfer device for the inoculation of culture bottles would decrease the risk of BC contamination. Regarding other causes of BC contamination, blood collection via an intravascular line was performed in only one out of 40 BC contamination events, possibly because the rate of central line use in this study was approximately half that reported in nationwide surveillance (0.047 vs. 0.09/patient days) [22].

In the control period, a particularly high contamination rate was observed in August and September. During these 2 months, *B. cereus* (five strains) and *B. subtilis* (three strains) were isolated as contaminants, and a significantly higher contamination rate of *Bacillus* spp. was found in these months than in other months. Additionally, true bacteremia caused by *Bacillus* sp. was found in four events (*B. cereus*, three strains) in the summer season compared with being found in three events in other seasons during the control period. These findings suggest an increase in skin colonization or environmental contamination by *B. cereus* during the summer.

Ashkenazi-Hoffnung et al. [23] reported a 2.5-fold increase in the number of *Bacillus* isolates from August to October compared with that in other months. In another study, the mean monthly BC rate rose from 25 to a peak of 75 *Bacillus*-positive BCs/10,000 BCs performed during June to August [24]. In a case–control investigation, *Bacillus*-positive BCs involved contamination or device colonization rather than infection in three-quarters of patients. Kobayashi et al. [25] found that detection of *B. cereus* was highest in summer. Therefore, the seasonality of *Bacillus* spp. isolated from BCs should be taken into account when comparing the incidence of BC contamination. With introduction of the bundle intervention, a seasonal increase in the contamination rates was prevented. This raises the question of whether the intervention's effectiveness might vary across seasons. However, the BC contamination rate in

seasons other than summer also decreased significantly from 1.61% in the control period to 0.93% in the bundle period, and the largest reduction was observed in CNS.

A new technology has been developed for collecting blood samples for BC. Several studies have shown that a further reduction in contamination may be achieved by discarding the initial blood draw portion, which potentially contains skin microorganisms [26]. A diversion device sequesters the initial 1.5–2.0 mL of blood, and samples are collected through an independent second flow path, creating a closed vein-to-bottle collection system. Venipuncture using the diversion device is associated with a considerable reduction in BC contamination compared with the standard blood collection procedure [27].

There are several limitations to this study. First, we did not evaluate the institutional use of chlorhexidine alcohol swab sticks or sterile blood transfer devices, which were used only for skin preparation in the collection of blood samples for BC or inoculation of the culture bottle, respectively. Second, background information such as comorbidities or antibiotic use for patients in the two study groups was not collected. It would be beneficial for assessing the applicability of the findings to different patient groups. Third, we did not evaluate adherence to elements of the protocol before blood collection, such as the use of sterile gloves. Finally, this was a retrospective, single-center study; thus, the findings may not be generalizable to broader patient populations.

In conclusion, implementation of our bundled blood collection intervention, including skin preparation with 1% chlorhexidine alcohol, disinfection of culture bottles, and use of a sterile blood transfer device, decreased BC contamination. Using a bundled approach, we achieved a low contamination rate of 1.0%, a novel target attainment. As an alternative to the two-needle technique, the effect of using a sterile blood transfer device should be investigated in prospective comparative clinical studies. Improved understanding of a variety of issues is required for best practices during the procedures involved in collecting and handling blood samples for BC.

## Supporting information

**S1 Table. Protocol for blood culture during the control period.**
(TIF)

**S2 Table. Swab cultures around the hole of a needle-removing device.**
(TIF)

**S3 Table. Reasons for the determination of blood culture contamination for each organism.**
(TIF)

**S4 Table. Comparison of the proportion of each organism among contaminants between the control and bundle periods.**
(TIF)

**S1 Fig.** (A) Blood is collected into the syringe (A-1). Then, the butterfly needle is removed from the syringe and blood in the syringe is transferred to the culture bottle using a blood transfer device (A-2). (B) Blood transfer device used to transfer blood to a culture bottle during venipuncture, a method prohibited in Japan.
(TIF)

**S1 Data.**
(XLSX)

## Acknowledgments

We thank Analisa Avila, MPH, ELS, of Edanz (https://jp.edanz.com/ac) for editing a draft of this manuscript.

## Author Contributions

**Conceptualization:** Yoshio Takesue.

**Data curation:** Miyuki Makino, Yousuke Kihara, Akiko Tanikawa, Hitoshi Ogashiwa, Yukiko Nakano.

**Formal analysis:** Takashi Ueda.

**Investigation:** Miki Doi.

**Methodology:** Yoshio Takesue.

**Project administration:** Yasushi Murakami, Soichiro Nakama.

**Supervision:** Kazuhiko Nakajima.

**Validation:** Yasuhiro Nozaki.

**Writing – original draft:** Miki Doi.

**Writing – review & editing:** Yoshio Takesue.

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
