## [Decision Letter · Decision Letter 0]

4 Sep 2024

PONE-D-24-32717Effect of blood culture collection bundles on decreasing the contamination ratePLOS ONE

Dear Dr. Takesue

Thank you for submitting your manuscript to PLOS ONE. After careful consideration, we feel that it has merit but does not fully meet PLOS ONE’s publication criteria as it currently stands. Therefore, we invite you to submit a revised version of the manuscript that addresses the points raised during the review process.

We look forward to receiving your revised manuscript.

Kind regards,

Aijaz Ahmad, Ph.D.

Academic Editor

PLOS ONE

Journal Requirements: When submitting your revision, we need you to address these additional requirements. 1. Please ensure that your manuscript meets PLOS ONE's style requirements, including those for file naming. The PLOS ONE style templates can be found at https://journals.plos.org/plosone/s/file?id=wjVg/PLOSOne_formatting_sample_main_body.pdf and https://journals.plos.org/plosone/s/file?id=ba62/PLOSOne_formatting_sample_title_authors_affiliations.pdf 2. Thank you for stating the following in your Competing Interests section: "The author(s) received no specific funding for this work." Please complete your Competing Interests on the online submission form to state any Competing Interests. If you have no competing interests, please state ""The authors have declared that no competing interests exist."", as detailed online in our guide for authors at http://journals.plos.org/plosone/s/submit-now   This information should be included in your cover letter; we will change the online submission form on your behalf.  

Reviewers' comments:

Reviewer's Responses to Questions

**Comments to the Author**

1. Is the manuscript technically sound, and do the data support the conclusions?

Reviewer #1: Partly

Reviewer #2: Yes

2. Has the statistical analysis been performed appropriately and rigorously? 

Reviewer #1: Yes

Reviewer #2: Yes

3. Have the authors made all data underlying the findings in their manuscript fully available?

Reviewer #1: Yes

Reviewer #2: Yes

4. Is the manuscript presented in an intelligible fashion and written in standard English?

Reviewer #1: Yes

Reviewer #2: No

5. Review Comments to the Author

Reviewer #1: The manuscript by Doi et.al. aimed to reduce blood culture contamination rates by implementation of a bundled approach to blood collection. They bundle included skin preparation with chlorhexidine-alcohol, disinfection of culture bottles, and use of a sterile blood transfer device. The authors found that the contamination rate was significantly decreased from 2.0% during the control period to 1.0% during the bundle period. This reduced, especially for coagulase-negative staphylococci. Additionally, the authors observed a seasonal spike in contamination rates, particularly for Bacillus species in August and September during the control period, although not seen in bundle period. The authors concluded that a bundled approach significantly decreased the blood culture contamination rate from 2.0% to 1.0%, which might have significant medical importance for future approaches. However, the study lacks some major experiments and explanation, as following:

Major comments:

1. In the study, the authors discussed the effect of blood collection via intravascular lines on BC contamination only during the bundle period, but no data was provided in the control period. I am unsure how the authors concluded that bundle approach reduced the contamination, when there is no baseline or control group they have to compare with.

2. The study showed a seasonal variation in contamination rates, particularly for Bacillus species, which was reduced in bundle approach. This suggests that the overall contamination rate might be influenced by external factors, like environmental factors, seasonal changes beyond the intervention. The authors have not mentioned whether seasonality has any influences on any specific types of bacteria. This raises the question of whether the interventions' effectiveness might vary across seasons. Further investigation is needed to understand this relationship.

3. In this study institutional use of chlorhexidine-alcohol and sterile blood transfer devices not evaluated. Additionally, the study also did not assess their direct impact on BC contamination rates. This limit understanding of how consistently these materials were used. The authors should include this information for better understanding of the reader.

Minor comments:

1. 1.Although the study mentioned about the potential sources of contamination, a detailed data should be provided on the frequency or types of contaminants identified. This information would be valuable in understanding the overall contamination picture.

2. The authors should include information on the specific characteristics of the study population (such as age, underlying conditions, or antibiotic use), as it would be beneficial for assessing the applicability of the findings to different patient groups.

Reviewer #2: The manuscript submitted by Doi et al investigated the effect of a specific blood collection bundle on reducing the contamination rate that may translate into correct diagnosis and better clinical outcomes for the patients. The manuscript collected data from a Japanese hospital over a period of 3 years with proper control period and concluded that the bundle approach did significantly reduced the contamination incidence well into an acceptable range. The paper shall be accepted after following minor revisions.

1. The English language and expression needs proper review from a native English speaker.

2. The materials and methods section shall be structured into sub sections with proper headings for each section.

3. The figures, tables and figure legends be presented according to the journal's guidelines.

6. PLOS authors have the option to publish the peer review history of their article (what does this mean?). If published, this will include your full peer review and any attached files.

Reviewer #1: No

Reviewer #2: No

---

## [Author Response · Author response to Decision Letter 0]

25 Sep 2024

We wish to express our appreciation to the Reviewer for the insightful comments on our paper.

Reviewer #1: 

Major comments:

1. In the study, the authors discussed the effect of blood collection via intravascular lines on BC contamination only during the bundle period, but no data was provided in the control period. I am unsure how the authors concluded that bundle approach reduced the contamination, when there is no baseline or control group they have to compare with.

Response: Because blood collection through intravascular line is one of the increased risk factors for BC contamination, blood draw from line was evaluated in the control and the bundle period.

Line 288-293: Seventeen of 109 patients with BC contamination in the control period and 5 of 40 patients with BC contamination in the bundle period had a central venous catheter when venipuncture was conducted for BC; blood was collected through the line in 0 of 17 patients with contaminated BCs during the control period and in 1 of 5 patients with contaminated BCs during the bundle period. During both periods, no blood was drawn through an arterial line in any patients with BC contamination.

2. The study showed a seasonal variation in contamination rates, particularly for Bacillus species, which was reduced in bundle approach. This suggests that the overall contamination rate might be influenced by external factors, like environmental factors, seasonal changes beyond the intervention. The authors have not mentioned whether seasonality has any influences on any specific types of bacteria. This raises the question of whether the interventions' effectiveness might vary across seasons. Further investigation is needed to understand this relationship.

Thank you for the valuable comment to improve the quality of the paper. According to the reviewer comment, Table 5. (Comparison of the contaminant isolation rate between the control and bundle periods in August and September and in the remaining 10 months.) was added. 

Line 325-329: The effectiveness of the bundle intervention was demonstrated not only in these 2 months but also in the other months, and there was a significantly lower BC contamination rate in the bundle period compared with the rate in the control period during the 10 months from October to July (0.93% vs. 1.64 %, p=0.007) (Table 5).

3. In this study institutional use of chlorhexidine-alcohol and sterile blood transfer devices not evaluated. Additionally, the study also did not assess their direct impact on BC contamination rates. This limit understanding of how consistently these materials were used. The authors should include this information for better understanding of the reader.

Response: We agree with the reviewer comments, and we demonstrated that key elements of the bundle to limit the contamination (The chlorhexidine-alcohol swab and the sterile blood transfer device) were not conducted in the control period 

The chlorhexidine-alcohol swab

Fig 1.

Line 240-243: In the control period, a povidone-iodine swab stick (Povidone Plus®, Kawamoto Corporation, Osaka, Japan) was exclusively used for skin preparation before BC blood collection. Chlorhexidine alcohol swabs were introduced for the first time in May 2022 for bundle implementation (Fig 1.).

Sterile blood transfer devices

Line 267-275: Sterile blood transfer devices were introduced in December 2021 to transfer venous blood from a syringe to an evacuated blood collection tube for biochemical and peripheral blood cell examination. During the control period, use of the device was prohibited for transferring blood from a syringe to a BC bottle, and a conventional single-needle transfer technique was adopted. Use of the device to transfer blood to culture bottles was started following the introduction of the bundle, and institutional median monthly use significantly increased to 585 kits (IQR 424.5–646.5) in the bundle period (May 2022 to April 2024) comparing with that in the previous 6 months (p=0.008) (Fig 2).

Minor comments:

1. 1.Although the study mentioned about the potential sources of contamination, a detailed data should be provided on the frequency or types of contaminants identified. This information would be valuable in understanding the overall contamination picture.

Response: Detailed data was provided on the frequency or types of contaminants identified in Table 4. (Comparison of the blood culture contamination rate (per set) for each organism between the control and bundle periods) 

2. The authors should include information on the specific characteristics of the study population (such as age, underlying conditions, or antibiotic use), as it would be beneficial for assessing the applicability of the findings to different patient groups.

Response: The comparison of the background in study group is needed. Table 1 was modified and median age, gender and No. of set obtained in emergency outpatient clinic (rate) was added. However underlying condition or antibiotic use in 2 study groups could not be evaluated. This was mentioned in the limitation of the study.

Line 415-418: Second, background information such as comorbidities or antibiotic use for patients in the two study groups was not collected. It would be beneficial for assessing the applicability of the findings to different patient groups.

Reviewer #2: 

1. The English language and expression needs proper review from a native English speaker.

Response: Paper was corrected by a native English speaker

Line 431-433: Acknowledgment

We thank Analisa Avila, MPH, ELS, of Edanz (https://jp.edanz.com/ac) for editing a draft of this manuscript.

2. The materials and methods section shall be structured into sub sections with proper headings for each section

Response: Subsection with headings was prepared in the material and methods. 

Study design; method of bacterial culture; definition of BC contamination; the bundle intervention for collecting blood samples for BC; adherence for the BC bundle; Evaluation for the BC contamination rate; and Statistical analyses

3. The figures, tables and figure legends be presented according to the journal's guidelines.

Base on the reviewer comment, the figures, tables and figure legends were corrected according to the journal’s guidelines

---

## [Decision Letter · Decision Letter 1]

14 Nov 2024

Effect of blood culture collection bundles on decreasing the contamination rate

PONE-D-24-32717R1

Dear Dr. Takesue 

We’re pleased to inform you that your manuscript has been judged scientifically suitable for publication and will be formally accepted for publication once it meets all outstanding technical requirements.

Kind regards,

Aijaz Ahmad, Ph.D.

Academic Editor

PLOS ONE

Reviewers' comments:

Reviewer's Responses to Questions

**Comments to the Author**

1. If the authors have adequately addressed your comments raised in a previous round of review and you feel that this manuscript is now acceptable for publication, you may indicate that here to bypass the “Comments to the Author” section, enter your conflict of interest statement in the “Confidential to Editor” section, and submit your "Accept" recommendation.

Reviewer #1: All comments have been addressed

Reviewer #2: All comments have been addressed

2. Is the manuscript technically sound, and do the data support the conclusions?

Reviewer #1: Yes

Reviewer #2: Yes

3. Has the statistical analysis been performed appropriately and rigorously? 

Reviewer #1: Yes

Reviewer #2: Yes

4. Have the authors made all data underlying the findings in their manuscript fully available?

Reviewer #1: Yes

Reviewer #2: Yes

5. Is the manuscript presented in an intelligible fashion and written in standard English?

Reviewer #1: Yes

Reviewer #2: Yes

6. Review Comments to the Author

Reviewer #1: The authors have addressed all the comments from the reviewers. They re-oriented and added several tables/figures as per reviewers comments to make the manuscript easy to understand for the readers.

Reviewer #2: (No Response)

7. PLOS authors have the option to publish the peer review history of their article (what does this mean?). If published, this will include your full peer review and any attached files.

Reviewer #1: No

Reviewer #2: No

---

## [Editor Report · Acceptance letter]

17 Dec 2024

PONE-D-24-32717R1 

PLOS ONE

Dear Dr. Takesue, 

I'm pleased to inform you that your manuscript has been deemed suitable for publication in PLOS ONE. Congratulations! Your manuscript is now being handed over to our production team.

Kind regards, 

on behalf of

Dr. Aijaz Ahmad 

Academic Editor

PLOS ONE